# Radio over Fiber: An Alternative Broadband Network Technology for IoT

**Diego F. Paredes-Páliz** [1,*] , **Guillermo Royo** [1] , **Francisco Aznar** [2] , **Concepción Aldea** [1] **and Santiago Celma** [1]

1   Electronic Design Group—Aragón Institute of Engineering Research, Universidad de Zaragoza, 50009 Zaragoza, Spain; royo@unizar.es (G.R.); caldea@unizar.es (C.A.); scelma@unizar.es (S.C.)

2   Electronic Design Group—Aragón Institute of Engineering Research, Centro Universitario de la Defensa, 50090 Zaragoza, Spain; faznar@unizar.es

\*   Correspondence: dparedes@unizar.es

**Abstract:** Wireless broadband access networks have been positioning themselves as a good solution for manufacturers and users of IoT (internet of things) devices, due mainly to the high data transfer rate required over terminal devices without restriction of information format. In this work, a review of two Radio over Fiber strategies is presented. Both have excellent performance and even offer the possibility to extend wireless area coverage where mobile networks do not reach or the 802.11 network presents issues. Radio Frequency over Fiber (RFoF) and intermediate Frequency over Fiber (IFoF) are two transmission strategies compatible with the required new broadband services and both play a key role in the design of the next generation integrated optical–wireless networks, such as 5G and Satcom networks, including on RAU (Remote Antenna Unit) new functionalities to improve their physical dimensions, employing a microelectronic layout over nanometric technologies.

**Keywords:** IEEE 802.11; intermediate frequency over fiber; internet of things; radio over fiber; wireless access networks

---

## 1. Introduction

The upcoming internet of things (IoT) technological frontier arouses enormous interest. Essentially, its deployment potential depends on the available wireless connectivity, and thus several attempts have been made to adapt new IoT applications to various existing wireless solutions, including more recent projections for new mobile networks developed to be employed in the Industry 5.0 era [1]. The convergence of wireless communications and fiber optic systems has emerged as a promising solution to support the rate of growth for data traffic demand in wireless applications. However, there is a general consensus that due to the enormous amount of potential IoT applications (home automation, security, wearable, sensor networks, industrial automation, precision agriculture, remote surveillance, or the so-called Smart City), there is no single standard capable of satisfying the particular needs of each one, in terms of complexity and cost, power consumption and transmission speed [2,3].

At the moment, it is still unclear which technologies will take over to meet the increased connectivity demand in the long run, allowing a huge bandwidth consumption of future 5G networks that are already being deployed [4]. Research is underway in different areas because possible solutions will require the concurrence of various technologies [5]. For example, an obvious solution given by radio frequency spectrum saturation is to expand its use up to millimeter waves (30 to 300 GHz), as those frequency bands currently do not support many telecommunications services and offer channels with a huge capacity. A goal for IoT networks is to take advantage from non-assigned frequencies in the middle of a frequency band to connect IoT devices each other.

As an example, Non-Orthogonal Multiple Access (NOMA) has been tested to improve spectral efficiency for new wireless standard communications, in particular with new IoT applications where NOMA-based Resource Allocation allows multiple nodes to share the same spectrum [6]. However, a tremendous effort must be made to design and manufacture communication systems operating at such frequencies. Moreover, at high frequencies attenuation increases, then techniques such as beamforming, which consists of radiating with high directivity, are employed, and in this way the reachable distance for millimeter waves transmission is increased [7].

## 2. Technological Approach

In new wireless access networks, there is an option to avoid the attenuation issue through distributed antenna systems (DAS) implementation. That is, a high-density network composed of small base stations (small cells) distributed by urban areas or industrial environments to provide good coverage over a specific area, as is shown in Figure 1. These cells, operating at millimeter waves, will need antennas of reduced size compared to current ones, which will simplify their deployment. On the other hand, reduced antenna size brings up the possibility that future base stations will be able to operate using massive multiple input–multiple output (MIMO) configurations to strongly increase the maximum data capacity, if individual channels are multiplexed over the same communications infrastructure.

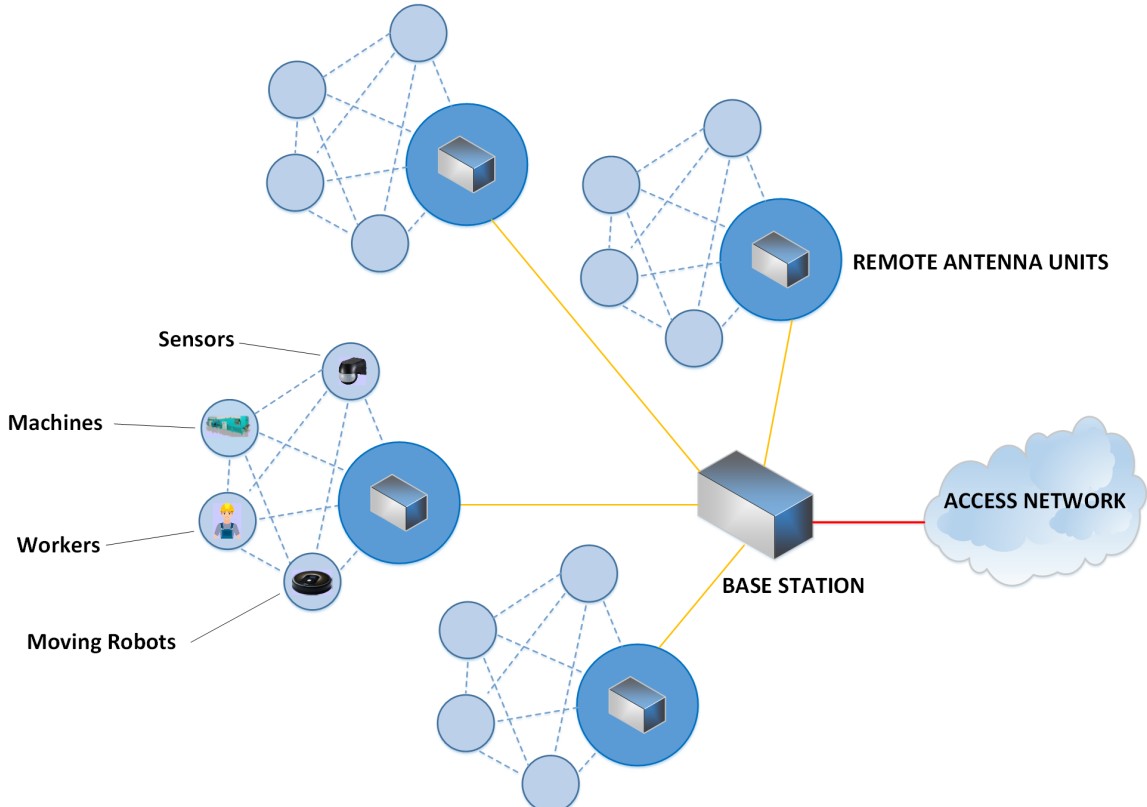

**Figure 1.** Scheme of a distributed antenna system.

A new telecommunications scenario is still being set, with fewer doubts that over most commonly used standards for IoT implementation (WiFi, Bluetooth, ZigBee, Thread, SigFox, NFC, etc.), the WiFi standard is the preferred option and has a greater development potential [2]. In fact, there are many predictions that coincide in pointing out that the role played by the WiFi standard in IoT systems, relevant already, will become more important with time and will establish as the dominant option in the mid-term [8,9].

From its appearance in 1997, the IEEE 802.11 standard has evolved to become a family of standards operating in the 2.4 GHz (802.11b/g/n) and 5 GHz (802.11a/n/ac) bands, offering data transmission rates that exceed 1 Gbps (802.11ac) with MIMO systems.

This versatility is currently driving the research to new versions within the family of WiFi standard in order to adapt it to IoT networks, while maintaining a common framework that allows interoperability and simplifies design cycles. The industry forecasts [10] a progressive implementation of the 5 GHz (802.11ac) and 6 GHz (802.11ax) standards until 2021 and subsequently a coexistence with other 60 GHz WiFi versions of progressive implementation from 2024.

## 3. Wireless Broadband Integration

The RF spectrum in Wi-Fi frequency bands is tending towards a constant saturation due to the increasing demand and density of users and networks. In this scenario, low power access nodes operating at 5 GHz band is an ideal approach to overcome this limitation. The area coverage of these nodes is much smaller, so that very few networks concur in the same area. This coverage reduction is compensated by a greater deployment of access points.

In other words, instead of using 2.4 or 5 GHz base stations with maximum coverage, which is a source of interference in high-density networks of IoT devices, the DAS approach defines a considerable low coverage base stations number, mainly working at 5 GHz.

Therefore, they show less susceptibility to interference, present at densely populated areas, where a large number of wireless networks concur and it can be very difficult to achieve high data transmission rate.

In this context, the convergence of wireless communications and fiber optic systems in a DAS can become a promising technique to provide broadband wireless access services, in a range of applications that support the growing wireless data traffic, either indoors or outdoors, combining the best of both technologies: wireless systems mobility and low attenuation combined with high capacity from optical fibers, in a world where high-speed connectivity is required anytime and anywhere [11–14].

In the last years, mixed fiber–wireless communication DAS fed by multi-mode fibers (MMF) are gaining space as the most promising solution to achieve efficient, cost-effective, and high-capacity transmissions in short range communications [15].

These systems are flexible and there is a good compromise between data transmission capacity, accessibility, and the overall cost of installation and maintenance, allowing a good convergence of optical fiber capacity and wireless access flexibility. In this approach, the signal is generated and processed in a base station (BS) and is distributed through MMF to several remote antenna units (RAU), which provide an optical–wireless interface as is shown in Figure 2 [13].

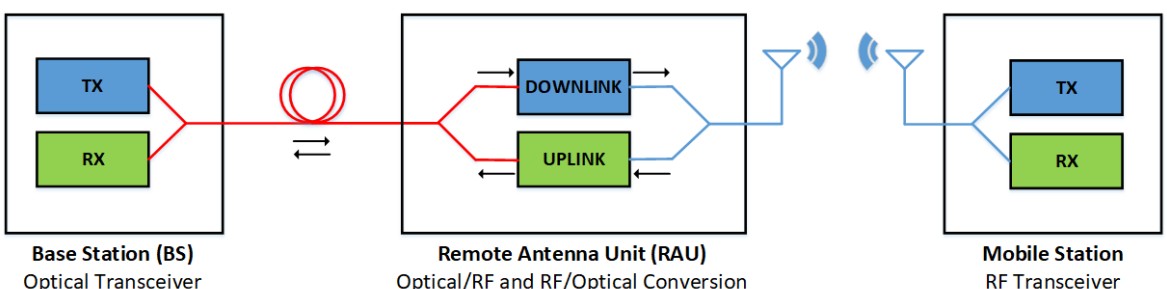

**Figure 2.** Conceptual scheme of a mixed fiber–wireless communication system based on distributed remote antenna units.

## 4. RAU Operation Scheme

In communications systems, information is transmitted over a transmission medium, being a wireless channel or a physical medium; in RAU, an uplink and downlink wireless channel is enabled, between the BS and user device, configuring a bidirectional channel. On the other hand,

RAUs downlink receives the optical signal from the BS, meanwhile uplink transmits another in the opposite way, back to the BS.

There is a considerable difference with respect to the traditional wireless networks that consist of broadcasting the analog signal information from a single transmitter, in this approach the DAS configuration offers much better coverage and data transmission capacity with a decrease in interference with other wireless networks [14].

A dense RAU devices deployment means that the key of a DAS is to employ cost-effective components in the three main elements of the communication system. This is achieved with the use of MMF (Multimode fiber), which offers electromagnetic isolation and a large capacity, along with cost-effective photonic devices, such as vertical-cavity surface-emitting lasers (VCSEL). Lastly, a low-cost, low-power, fully integrated RAU must be designed in cost-effective technologies such as CMOS and a moderate complexity to reduce the cost of DAS. An operation scheme of an electrical/optical (EO) and optical/electrical (OE) RAU domain converter is presented in Figure 3, with a description of main blocks and key parameters.

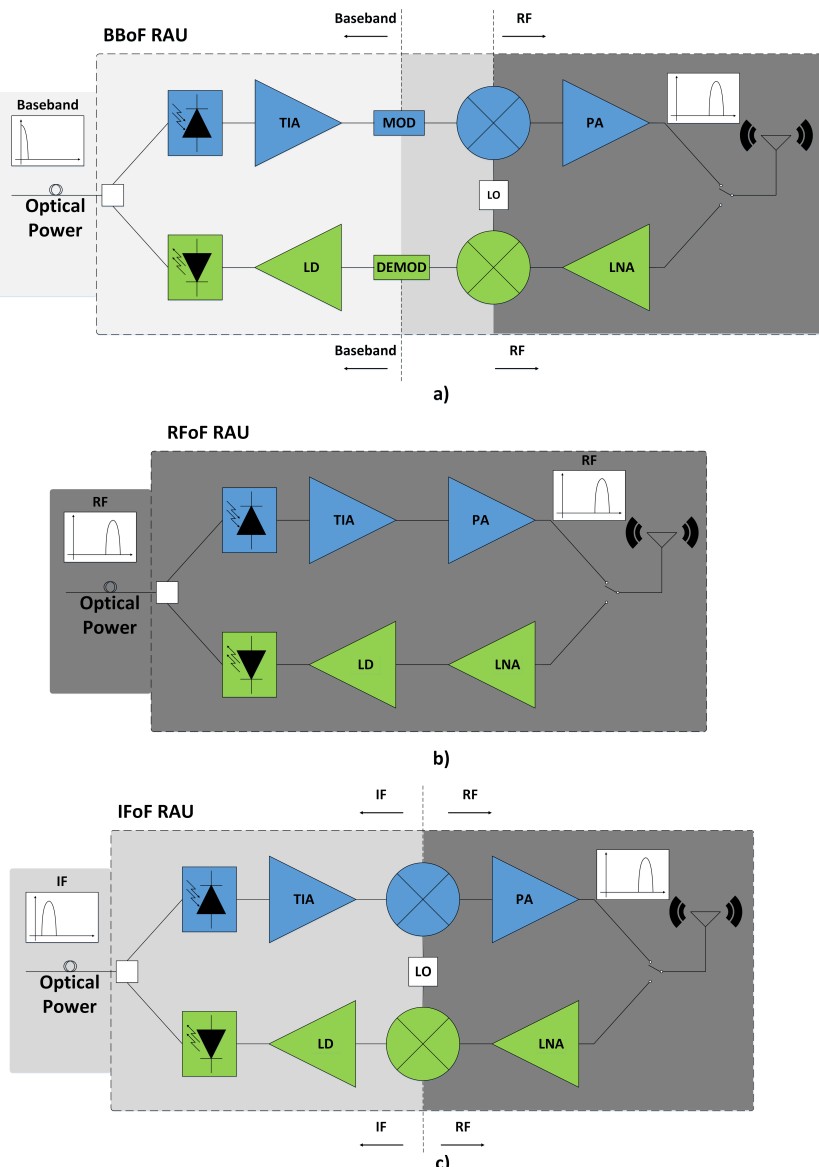

**Figure 3.** Conceptual scheme of remote antenna units for: (**a**) Baseband over fiber (BBoF). (**b**) Radio frequency over fiber (RFoF). (**c**) Intermediate frequency over fiber (IFoF).

## 5. RAU Architectures

Mixed fiber–wireless systems offer several optical data transport configurations, which can be classified in three categories: baseband over fiber (BBoF), RF over fiber (RFoF) and intermediate frequency over fiber (IFoF). These schemes are shown in Figure 3 and they are named upon how the data transmission through the fiber is performed.

### 5.1. Baseband over Fiber

The BBoF scheme follows the typical configuration of optical communication systems for long-reach applications. With this scheme, very high bit rates in the order of several Gbps can be achieved. However, this scheme requires complex RAU configuration and design, as it must perform data modulation and demodulation, as well as frequency conversion, also increasing power consumption. Since a high number of RAUs is needed, the overall cost of a BBoF-based DAS can rise considerably, therefore this approach is precluded on many scenarios and serves only for home area networks.

### 5.2. Radio Frequency over Fiber

On the opposite side, the RFoF scheme, called also in some scientific literature RoF, presents the simplest RAU architecture, Figure 3b, because the RAU only has to perform opto/electrical (O/E) and electro/optical (E/O) conversion and signal amplification. It is the most flexible RAU, since as it works in the RF domain, it is absolutely transparent to the data encoding or communication standard.

The RF signal is generated neither at BS nor at RAU, so as not to perform any process: modulation–demodulation even frequency conversion. In this scenario, systems require much higher linearity than BBoF to transmit RF signal properly and present distance limitations when it is proposed to transmit on different wireless standards at higher RF frequencies, due to the high fiber chromatic dispersion at such frequencies [16]. Some attempts to design RAUs for RFoF have been made in recent years with promising results and performance, transmitting RF at 2.5 GHz [17,18], 5 GHz [19], or 12 GHz [20].

One of the main disadvantages of RFoF systems is that they require high-speed circuit design and high-performance photonic devices, therefore increasing the power consumption and the overall cost of the DAS significantly. Moreover, at higher frequencies [21], there are undesired effects such as power penalty, fading, and nonlinearities, that induce spectrum broadening of baseband data, around the carrier signal [15,22–26].

### 5.3. Intermediate Frequency over Fiber

Half way between BBoF and RFoF we find IFoF. These systems present advantages over both BBoF and RFoF. As in the RFoF approach, RAUs for IFoF do not require the implementation of a modulator/demodulator. Therefore, the complexity in the design of the RAU is much simpler than that in BBoF systems, and the power consumption is significantly lower, Figure 3c.

The signal is generated at the BS with the same modulation format as the RF signal, but at a lower frequency [15,26,27]. Therefore, the flexibility of IFoF RAUs is just a step below RFoF as it is almost transparent to the communication standard, with the only need of tuning the RF carrier frequency. Therefore, the complexity of the RAU increases with an IFoF scheme with respect to an RFoF system.

To recover the original RF signal, the RAU now requires a high-speed mixer and a stable local oscillator (LO), which is used for both downlink and uplink, with an accurately tuned frequency to carry out up-conversion and down-conversion, respectively. Furthermore, to generate the LO signal, a pilot carrier can be delivered to the RAU optically, acting as external reference, allowing a higher flexibility and frequency tuning without the need of quartz crystal or similar devices.

With all this, IFoF presents major advantages over RFoF. First, since the optical signal is modulated with a much lower frequency, the use of an IFoF system minimizes the effect of MMF chromatic

dispersion and simplifies significantly the integrated circuit (IC) design [28,29]. Moreover, it also drops the overall cost of the system, since lower performance and less expensive photonic devices can be used at both the BS and RAUs, for both O/E conversions in the downlink and E/O conversion in the uplink.

## 6. Discussion

Wireless communications development is facing a huge upcoming challenge. The immense growth of data demand over recent years is showing that the capacity of currently deployed wireless access networks must increase considerably in the following years. Otherwise, it will not be possible to satisfy the future needs, which will definitely slow down our development and will probably have a negative impact on the global economy.

Several approaches are tending towards mixed fiber–wireless system configurations, which are the most promising solution to deal with the enormous increase of data demand in the near future. Nevertheless, each scenario and application will present specific requirements, so that the preferred solution may vary between the available DAS and RAU structures. Therefore, scenarios with a very high bitrate need will probably choose the BBoF scheme, despite its higher costs. On the contrary, in applications where a high system flexibility and low cost are mandatory, the most preferable approaches will be RoF and IFoF, while the choice will depend on several factors, such as RF frequency or the maximum budget in each case.

This paper shows the main alternatives to build a broadband network using a combination of fiber and wireless technologies, discarding BBoF due to their constraints described above. Summarizing all of this, Table 1 presents some specifications for RFoF and IFoF that can be considered in future works.

**Table 1.** A summary for RFoF and IFoF architectures reviewed in this work.

| RoF Type | RF/IF (GHz) | Fiber | Signal Type | Light Source | Photo Detector | Technology | Link | Application | Ref. Year |
|---|---|---|---|---|---|---|---|---|---|
| RFoF | 5/- | MMF | 64-QAM 54 Mbps | 850-nm | APD | 0.180-μm CMOS | Downlink | Indoor RAU | [19] 2013 |
| RFoF | 60/- | MMF | BPSK 1.6 Gbps | 850-nm | APD | 0.25-μm SiGe BiCMOS | Downlink | Indoor RAU | [21] 2012 |
| RFoF | 40/- | SMF | SSB 7.8 Gbps | N/A | N/A | SiPh-SOI | Downlink | RRF | [26] 2019 |
| IFoF | 28/1.7–2.7 | SMF | 64-QAM 1.5 Gbps | 1550-nm 1310-nm | N/A | Commercial ICs | Fronthaul | Outdoor RAU | [27] 2018 |
| IFoF | 2.1/0.04 | MMF | 16-QAM N/A | VCSEL | APD | 65-nm CMOS | Fronthaul | Indoor RAU | [28] 2017 |
| IFoF | 5/0.1–0.3 | MMF | 64-QAM 54 Mbps | 1550-nm | PIN | 65-nm CMOS | Downlink | Indoor RAU | [29] 2019 |

RRF = Remote Radio Frontend, APD = Avalanche Photodiode, PIN = p-i-n Photodiode.

In this table are presented the results obtained in previous experimental works, highlighting specific design parameters for both RFoF and IFoF; this provides a clearer picture of how the two RoF strategies operate in terms of data transmission rate, source light (wavelenght), frequencies operation, applications, and type of fiber. The IFoF scheme presents a potentially good option for design, fabrication, and characterization for RAU devices, offering a cost-effective choice for DAS deployment [29,30].

The motivation to choose the IFoF scheme for the complete RAU design over the other configurations is driven by the fact that this approach is considered the main candidate to develop high data capacity wireless networks in densely populated areas. These future networks need to be deployed with a very high number of short-reach nodes in a cost-efficient way, which is the reason why IFoF systems are generating an increasing interest over the last years [31,32].

RAU working with IFoF is a good option in terms of IC design with a practical consideration, and uses a single frequency and channel if the main purpose is to increase coverage area including those called dead zones to improve reliability.

A better way to understand the reason to choose a IFoF RAU is the possibility to escalate electronic design to multiplex several channels with different intermediate frequencies, increasing data rate capacity and enhancing spectral efficiency.

## 7. Conclusions

IFoF technology is the most promising solution for IoT network deployment due to workable electronic IC design and integration with currently available WiFi 802.11 and emerging WiFi 802.11ad/ay standards at 60 GHz. This can be the start point for future research to achieve a good RAU design and fabrication, that allow to measure the main parameters, in order to achieve good results employing monolithic technologies.

**Author Contributions:** All authors contributed to the information analysis and the writing of the paper. All authors have read and agreed to the published version of the manuscript.

**Funding:** This paper has been supported by MINECO-FEDER Project Grant (TEC2017-85867-R) and MICINN fellowship program to Diego Paredes (PRE2018-083578).

**Conflicts of Interest:** The authors declare no conflict of interest.

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
