# Peer review of "Radio over Fiber: An Alternative Broadband Network Technology for IoT"

_electronics, doi:10.3390/electronics9111785_

Round 1
Reviewer 1 Report
this work is presented a review of two Radio over Fiber strategies which have an excellent performance and even offer the possibility to extend wireless area coverage where mobile networks does not arrives or 802.11 network presents issues. The paper is very interesting. Some comments can be revised.
- The references in the introductions are not enough, there are only three conference in the introductions. The author can described some work on improving IoT performance. For example, in paper "NOMA-based Resource Allocation for Cluster-based Cognitive Industrial Internet of Things. IEEE Transactions on Industrial Informatics", It can be indicated in the introduction that "NOMA-based Resource Allocation was proposed for IoT by allowing multiple nodes to access the same spectrum, which can improve the IoT's throughput greatly."
- The working process of the system should be indicated in the paper.
- some simulation results should be given in the paper to varify the peformance of the designed system.
- How to get the table 1. which conclusion can we get from the table?
Reviewer 2 Report
The authors review three main mixed fiber-wireless systems for remote antenna unit architectures, namely, the baseband over fiber (BBoF), RF over fiber (RFoF) and intermediate frequency over fiber (IFoF) schemes, and point out that “IFoF systems are generating an increasing interest over the last years”. This review paper might be helpful for future deployment of high-dense and high-efficient access points of networks. However, my main concern is that the arguments made in this paper are based mostly in the literature around ten years from now. For example, from refs. [11] published in the year 2005, the authors claim that “In the last years, mixed fiber-wireless communication (DAS) fed by multi-mode fibers (MMF) are gaining space as the most promising solution to achieve efficient, cost effective and high capacity transmissions in short range communications [11]”. Is this statement still up to date for the rapid developing area of IoT as of today in 2020? In addition, only 25 reference papers are cited including one main reference (the last reference [25]) from the authors’ own work published last year. Therefore, I think more recent reference papers from other leading groups would better support this review paper, especially the evaluation statements for the overall cost of installation and maintenance for IoT.Author Response
Please see the attachment.

Round 2
Reviewer 1 Report
accept
Reviewer 2 Report
The authors have modified the original manuscript according to my comment -- they have add more recent references to support their arguments. So I recommend its publication in Electronics.